# Antagonistic Skin Toxicity of Co-Exposure to Physical Sunscreen Ingredients Zinc Oxide and Titanium Dioxide Nanoparticles

**DOI:** 10.3390/nano12162769

**Published:** 2022-08-12

**Authors:** Yan Liang, Aili Simaiti, Mingxuan Xu, Shenchong Lv, Hui Jiang, Xiaoxiang He, Yang Fan, Shaoxiong Zhu, Binyang Du, Wei Yang, Xiaolin Li, Peilin Yu

**Affiliations:** 1Department of Toxicology, and Department of Medical Oncology of Second Affiliated Hospital, Zhejiang University School of Medicine, Hangzhou 310058, China; 2Lishui International Travel Health-Care Center, Lishui 323000, China; 3MOE Key Laboratory of Macromolecular Synthesis and Functionalization, Department of Polymer Science & Engineering, Zhejiang University, Hangzhou 310027, China; 4Department of Biophysics, Zhejiang University School of Medicine, Hangzhou 310058, China; 5Technical Center of Animal, Plant and Food Inspection and Quarantine of Shanghai Customs, Shanghai 200135, China

**Keywords:** ZnO NPs, TiO_2_ NPs, antagonistic skin toxicity, EpiSkin

## Abstract

Being the main components of physical sunscreens, zinc oxide nanoparticles (ZnO NPs) and titanium dioxide nanoparticles (TiO_2_ NPs) are often used together in different brands of sunscreen products with different proportions. With the broad use of cosmetics containing these nanoparticles (NPs), concerns regarding their joint skin toxicity are becoming more and more prominent. In this study, the co-exposure of these two NPs in human-derived keratinocytes (HaCaT) and the in vitro reconstructed human epidermis (RHE) model EpiSkin was performed to verify their joint skin effect. The results showed that ZnO NPs significantly inhibited cell proliferation and caused deoxyribonucleic acid (DNA) damage in a dose-dependent manner to HaCaT cells, which could be rescued with co-exposure to TiO_2_ NPs. Further mechanism studies revealed that TiO_2_ NPs restricted the cellular uptake of both aggregated ZnO NPs and non-aggregated ZnO NPs and meanwhile decreased the dissociation of Zn^2+^ from ZnO NPs. The reduced intracellular Zn^2+^ ultimately made TiO_2_ NPs perform an antagonistic effect on the cytotoxicity caused by ZnO NPs. Furthermore, these joint skin effects induced by NP mixtures were validated on the epidermal model EpiSkin. Taken together, the results of the current research contribute new insights for understanding the dermal toxicity produced by co-exposure of different NPs and provide a valuable reference for the development of formulas for the secure application of ZnO NPs and TiO_2_ NPs in sunscreen products.

## 1. Introduction

With the growing awareness of adverse biological effects caused by solar ultraviolet (UV) radiation, sunscreen products are increasingly making their way into people’s everyday life [1], showing up in the form of oils, gels, lotions, mists, and even roll-on preparations, in addition to the traditional sun-creams [2]. According to the protective mechanism, sunscreen agents can be classified into two broad categories [3]. The physical (also named inorganic) sunscreens principally work by reflecting and scattering the UV radiation, while the chemical (also named organic) sunscreens absorb UV radiation and convert it into molecular kinetic energy or heat energy [4,5]. Generally, chemical sunscreen products containing active organic ingredients have a higher risk of causing skin-damaging effects [2], such as photoirritation, photosensitization, and contact dermatitis [6]. In comparison, pure physical sunblock agents based on inorganic UV filters are relatively mild and are often used for children and sensitive skin. However, traditional inorganic UV filters such as zinc oxide (ZnO) and titanium dioxide (TiO_2_) with large particles often inevitably lead to cosmetically unacceptable appearance, thus limiting the popularization of physical sunblock products to a large extent [7,8].

Starting in the early 1990s, zinc oxide nanoparticles (ZnO NPs) and titanium dioxide nanoparticles (TiO_2_ NPs) overcame the shortcoming of ghostly white lumps on the appearance caused by deposits of the initial minerals, making sunscreens and other cosmetics containing nanoparticles (NPs) popular among consumers [9]. Many governments, such as those of Europe, China, Australia, and the United States, have authorized ZnO NPs and TiO_2_ NPs as qualified UV filters at a concentration lower than 25% [6]. When the raw materials of UV filters were ground to the nanoscale, the physical shielding effects were not only limited to scattering and reflection, but also involve absorption of UV radiation. The absorption of UVB (290–320 nm) by TiO_2_ NPs was closely related to the particle size [10], showing that the smaller the particle size is, the shorter the peak absorption spectrum appears [8]. With a flat absorbance curve of UVA (320–400 nm) radiation [10], the critical wavelength of ZnO NPs was independent of its particle size [8]. Therefore, the primary particle sizes of TiO_2_ NPs used in sunscreens were often between 10 and 30 nm. For ZnO NPs, primary particle sizes from 10 to 200 nm were available, but mainly the grades with larger particles were used in sunscreen products [8,11]. To provide broad-spectrum protection against UVB and UVA radiation, the combination of ZnO NPs and TiO_2_ NPs was often applied in the sunscreen manufacturing field [6].

However, the broad usage of sunscreen products containing NPs has gradually raised concerns regarding their potential toxicity, and the controversy over the safety of NPs for human skin has never disappeared [12,13]. Major questions regarding the safety of ZnO NPs in sunscreens were related to their toxic potential in human umbilical vein endothelial cells (HUVECs), epidermal cells (A431), viable human keratinocytes, and fibroblasts, and it was demonstrated that ZnO NPs can induce cytotoxicity [11], oxidative stress [14] and DNA damage to skin [15]. Although TiO_2_ NPs have been reported to be less toxic than other metal oxide NPs [16,17], and several reports have even shown their biosafety in skin applications [18,19,20,21], there have been several reports revealing their toxic damage to cell viability, oxidative stress, and cellular inflammation [22,23]. However, under the common situation of NPs co-applications, these single exposure reports from nanobiotechnological and nanotoxicological studies have been considered unrepresentative for actual consumer use.

To date, a key unresolved issue with NPs in sunscreens is the dermal toxicity induced by co-exposure of both ZnO NPs and TiO_2_ NPs. Unlike the effects observed after exposure to individual NPs, the joint toxicity induced by co-exposure to different NPs is extremely complex [24]. For example, TiO_2_ NPs might show different antagonistic, partially additive, or synergistic toxicity due to their different proportion in the co-exposure system [25]. Nontoxic concentrations of ZnO NPs may be a toxic enhancer of other NPs in human cells [26]. Yu et al. found that co-existence with TiO_2_ NPs displayed a dose-dependent mitigation effect on the cytotoxicity of ZnO NPs to a bacterium, and many genes related to heavy metal scavenging, DNA repair, and oxidative stress were upregulated after the NP co-exposure, indicating that ZnO NPs and TiO_2_ NPs generated antagonistic effects on biological toxicity [27]. Conversely, Ogunsuyi et al. found that the mixture of these two NPs resulted in a significant synergistic effect on mice testis injury; the combination significantly reduced sperm motility and sperm number and markedly increased sperm abnormalities and aberrations compared with exposure to each NP type alone [28]. In addition, for human Jurkat cells, TiO_2_ NPs also increased the cytotoxicity of ZnO NPs and reduced the phosphorylation of the signaling proteins [29]. As can be seen, the health hazard induced by co-exposure to different NPs has gradually attracted people’s attention. However, the joint effects are still controversial, and relatively little research involving combined toxicity induced by these two NPs has been performed on skin cells or skin models.

Therefore, in this study, we paid attention to the HaCaT cells and EpiSkin models, aiming to explore the joint skin effects caused by co-exposure of ZnO NPs and TiO_2_ NPs. In addition, further investigation regarding the potential mechanism behind the combined toxicity has also been conducted. Ultimately, we hope to provide some suggestions for the proportion formula for the secure application of ZnO NPs and TiO_2_ NPs in sunscreens.

## 2. Materials and Methods

### 2.1. Particles and Exposure

ZnO NPs (particle size: 30 ± 10 nm, 99.9% metals basis) and TiO_2_ NPs (particle size: 30 nm, 99.8% metals basis) used in this study were purchased from Shanghai Aladdin Bio-Chem Technology Co. (Shanghai, China). The morphology and particle size of ZnO NPs and TiO_2_ NPs were examined by cryogenic transmission electron microscopy (Tecnai G2 Spirit 120Kv, FEI, Brno, Czech Republic). The hydrodynamic diameter and zeta potential of ZnO NPs and TiO_2_ NPs in different media (double-distilled water (ddH_2_O); DMEM high sugar medium with 10% fetal bovine serum (FBS)) were measured using dynamic light scattering (Nano-S90, Malvern Instruments, Malvern, UK) and Zetasizer Nano Series (Malvern Instruments, Malvern, UK), respectively. The hydrodynamic diameter and zeta potential were measured three times, and mean ± S.E.M (standard error of mean) was calculated.

The powders of ZnO NPs and TiO_2_ NPs were sterilized by autoclaving and suspended in double-distilled water (ddH_2_O) to obtain a stock solution of 4.0 mg/mL, then sonicated with an ultrasonic cell crusher (90% power, 30 min; Scientz08-III, Ningbo, China) before use. Different working concentrations of TiO_2_ NPs (0, 10, 30, 100, 200, and 300 μg/mL) and ZnO NPs (0, 5, 10, 20, 30, 50, and 100 μg/mL) were prepared by diluting the stock solutions with the same culture medium. To make the mixed suspension of these two NPs, one volume of x μg/mL TiO_2_ NPs and one volume of y μg/mL ZnO NPs were mixed together to finally produce two volumes of mixed suspension containing x/2 μg/mL TiO_2_ NPs with y/2 μg/mL ZnO NPs.

### 2.2. Preparation and Detection of FITC-ZnO NPs

According to the method described by Yung et al. [30], we applied fluorescein isothiocyanate (FITC, Aladdin Industry Co., Shanghai, China) to synthesize fluorescein-labeled ZnO NPs (FITC-ZnO NPs). Specifically, 4 mg ZnO NPs were dispersed into 3 mL dimethylformamide (DMF, Aladdin Industry Co., Shanghai, China) to form a ZnO NP suspension. Subsequently, 0.5 μL aminopropyltriethoxysilane (APTS, Aladdin Industry Co., Shanghai, China) solution was diluted in 25 μL DMF and then added to the ZnO NP suspension. Oxygen was eliminated by bubbling nitrogen through the suspension. Afterward, the suspension was sonicated at room temperature for 30 min. The modified ZnO NPs were then collected by centrifuging and washed with DMF 3 times. After washing, the ZnO NPs were resuspended into 0.5 mL DMF, followed by 30 min of sonication. FITC solution made up of 1 mg FITC and 0.5 mL DMF was mixed with the resuspension. The reaction suspension underwent vigorous stirring for 4 h. The resultant FITC-ZnO NPs were collected by centrifuging and washed with DMF thoroughly. Then, the FITC-ZnO NPs were dried under vacuum at 70 °C to remove excess solvent and stored as a powder for later use.

The fluorescence signal of FITC-ZnO NPs was examined using an ACEA NovoCyte Flow Cytometer (ACEA Biosciences, San Diego, CA, USA). A Fourier transform infrared spectrometer (FT-IR, Vector 22 Bruker, Karlsruhe, Germany) was employed to analyze the chemical structure of ZnO NPs and FITC-ZnO NPs. The dried NPs were mixed with KBr powders, and then the mixture was compressed into tablets for FT-IR measurement. The fluorescence spectrum was recorded using an FLS920 transient fluorescence spectrometer (Edinburgh Instruments Ltd., Livingston, UK) with a xenon lamp as the excitation source.

### 2.3. Cell Culture

The human immortalized keratinocyte cell line, HaCaT, was purchased from the cell bank of the Chinese Academy of Medical Sciences. Cells were cultured in DMEM high sugar medium (CR-12800, 313009, Cienry, Huzhou, China) supplemented with 10% FBS (P30-3306, PAN-Biotech, Aidenbach, German), 100 IU/mL penicillin, and 100 μg/mL streptomycin in a humidified incubator with 5% CO_2_ at 37 °C. Passages 3–30 were used in the experiments.

### 2.4. Cytotoxicity Assay

The viability of HaCaT cells was determined using Cell Counting Kit-8 (C0043, Beyotime, Shanghai, China) according to the manufacturer’s instructions. Briefly, cells were seeded in 96-well plates at a density of 1.5 × 10^4^ cells/well and treated with TiO_2_ NPs (0, 10, 30, 100, 200, and 300 μg/mL), ZnO NPs (0, 5, 10, 20, 30, 50, and 100 μg/mL), or their mixed suspensions with different combinations of doses, in the absence or presence of methyl-β-cyclodextrin (Mβ-CD, 2.5 mM, C4555, Sigma, St Louis, MO, USA) and ethylene diamine tetraacetic acid (CaEDTA, 0.5 mM, 60-00-4, Sigma, St Louis, MO, USA) for 3, 6, 12, or 24 h. After a certain time, cells were washed twice with phosphate-buffered saline (PBS) (B540626, Sangon Biotech, Shanghai, China) and CCK-8 was added to each well. After further incubation of 1.5 h, the absorbance at 450 nm was evaluated using a microplate reader (Tecan Infinite M200, Männedorf, Switzerland).

To observe the toxic effects caused by NPs on HaCaT cells more intuitively, the living/dead cells were labeled using calcein/PI (C2015M, Beyotime Biotechnology, Shanghai, China) dual staining and fluorescence imaging. Briefly, HaCaT cells were grown on glass-bottom dishes (D29-20-1.5P, Cellvis, Mountain View, CA, USA) to 70% confluence and treated with 20 μg/mL ZnO NPs, 100 μg/mL TiO_2_ NPs, or their mixtures for 24 h. After 30 min of dyeing in a dark incubator at 37 °C, cells were washed twice with PBS and visualized under a confocal fluorescence microscope (FV1000, Olympus, Tokyo, Japan). The data of fluorescence signals were analyzed using ImageJ (Rawak Software Inc., Stuttgart, Germany).

### 2.5. Comet Assay

HaCaT cells were firstly exposed 24 h to TiO_2_ NPs (100 μg/mL), ZnO NPs (20, 50, and 100 μg/mL), or the mixtures of different doses of ZnO NPs with 100 μg/mL TiO_2_ NPs. After exposure, the comet assay was performed according to the traditional protocol [31,32], cells were harvested, and slides were prepared according to the method described by Singh et al. [31] and modified by Bajpayee et al. [33]. The observation of the slides was performed on a BX61 fluorescence microscope (BX61, Olympus, Tokyo, Japan) equipped with a CCD camera and appropriate filter. The cellular DNA breaks were scored using the Opencomet analysis system provided with the ImageJ Software (Rawak Software Inc., Stuttgart, Germany). The percentage of DNA in tail (%DNA in tail, TD%) was applied for the final statistical analysis. Each experiment was performed at least three times.

### 2.6. Determination of Intracellular Zinc Contents

Cells were treated with ZnO NPs in the absence or presence of TiO_2_ NPs in 6-well plates for 6 h. After being collected and counted, the cells were pre-digested overnight with 0.5 mL concentrated nitric acid (HNO_3_) (216002104, YongHua, Changshu, China). Then, pre-digested products were heated for 30 min at 100 °C for thorough digestion. After cooling down, the samples were diluted with 2% (*v*/*v*) HNO_3_ to a total volume of 1 mL and filtered with the 0.22 μm water filtration membrane. Finally, 4 mL 2% (*v*/*v*) HNO_3_ was added to the filtered liquid to reach an acid concentration of less than 10%. Each sample was prepared in triplicate. The total zinc contents in samples were determined by atomic absorption spectroscopy (AAS, 240FS AA (flame), Agilent Technologies, Palo Alto, CA, USA) and finally averaged to each cell based on the cell count data.

### 2.7. Detection of Intracellular Zn^2+^

For intracellular Zn^2+^ ion detection, cells were treated with ZnO NPs (20 μg/mL) in the absence or presence of TiO_2_ NPs (100 μg/mL), CaEDTA, and Mβ-CD in 6-well plates for 6 h. Afterward, 1 mL FluoZin-3 (500 nM in culture medium, F24195, Invitrogen, Carlsbad, CA, USA) was added to each well and incubated for 30 min to probe the Zn^2+^ in the dark condition. Afterward, cells were washed three times with PBS and then 1 mL culture medium (without FluoZin-3) was added. After incubation for another 30 min, the cells were digested with 0.25% trypsin (CR-25200, 313009, Cienry, Huzhou, China) and resuspended using PBS into a flow tube to examine the fluorescence intensity with a NovoCyte Flow Cytometer.

### 2.8. Agglomeration State between ZnO NPs and TiO_2_ NPs

ZnO NPs, TiO_2_ NPs, and their mixed suspension (ZnO NPs:TiO_2_ NPs = 1:2) were prepared with ddH_2_O and sonicated using an ultrasonic cell crusher according to the protocol uniformly adopted in the whole experiment. Then, a tiny volume of NP suspension was dropped on the copper wire and left for drying for one night. The physical agglomeration state of ZnO NPs, TiO_2_ NPs, and their mixed suspension could be visualized under a scanning transmission electron microscope (STEM, Nova Nano 450, Thermo FEI, Brno, Czech Republic). To further quantify the agglomeration state of different NP suspensions, the hydrodynamic diameters of ZnO NPs, TiO_2_ NPs, and their mixed suspension were measured again using dynamic light scattering (Nano-S90, Malvern Instruments, Malvern, UK).

### 2.9. Cellular Uptake of ZnO NPs and TiO_2_ NPs

HaCaT cells were seeded in 6-well plates under the standard culture conditions. Before NP exposure, cells were pretreated with chlorpromazine hydrochloride (CPZ, 25 µM; 69-09-0, Sigma, St Louis, MO, USA), 5-(N-ethyl-N-isopropyl) amiloride (EIPA, 25 µM; 1154-25-2, MedChemExpress, Monmouth Junction, NJ, USA), and methyl-β-cyclodextrin (Mβ-CD, 2.5 mM; C4555, Sigma, St Louis, MO, USA) for 30 min at 37 °C. As reported [34], CPZ can specifically inhibit clathrin-mediated endocytosis, EIPA is an inhibitor of micropinocytosis, and Mβ-CD is a specific caveolin-mediated endocytosis inhibitor. Then, FITC-ZnO NPs, TiO_2_ NPs, or their mixtures were added into wells and incubated for another 6 h. Subsequently, cells were washed with PBS twice and collected for fluorescence signal analysis; that is, the uptake content of ZnO NPs was measured by fluorescence intensity of FITC using a NovoCyte Flow Cytometer, and the uptake content of TiO_2_ NPs was detected by titanium element detection using ICP-MS (7800, Agilent Technologies, Palo Alto, CA, USA).

### 2.10. The Detection of Dissociated Zn^2+^ from ZnO NPs

To detect the dissociated Zn^2+^ from ZnO NPs, the single suspensions of ZnO NPs (5, 20, 100 μg/mL) and the mixed suspensions of different dosages of ZnO NPs adding 100 μg/mL TiO_2_ NPs were prepared with ddH_2_O. The suspensions were then sonicated using an ultrasonic cell crusher as in the previous procedure. Then, the prepared suspensions were placed on a shaking table at 37 °C for 0.5, 6, and 24 h of standing and 1 h of shaking at 100 rpm. Solutions were then centrifuged at 20,000× *g* for 30 min to remove any NPs and precipitates. Afterward, a 0.4 mL supernatant was transferred to a 3.6 mL 2% high-purity nitric acid solution and analyzed by AAS (240FS AA (flame), Agilent Technologies, Palo Alto, CA, USA) to determine the total free Zn^2+^ in the solution.

### 2.11. Dermal Toxicity Assessment on Epidermal Model EpiSkin

With an area of 0.38 cm^2^, EpiSkin (S1, L’Oreal, Shanghai, China) is an in vitro reconstructed human epidermis (RHE) from normal human keratinocytes cultured on a collagen matrix at the air–liquid interface. Hematoxylin–eosin staining (HES) of paraffin sections from Shanghai EPISKIN Biotechnology Co., Ltd., showed that EpiSkin had the layered structures of the basal layer, spinous layer, stratum granulosum, transparent layer, and stratum corneum, which were very similar to normal human epidermis. As reported, the proportion of NPs allowed in sunscreens can be up to 25% [35], but around 20% usage was the common concentration of ZnO NPs in commercially available sunscreens [36,37]. To further verify the virtual toxic effect of NPs in sunscreens on human skin, based on an NP addition of 20% and recommended sunscreen usage of 2 mg/cm^2^, 0.4, 2.4, 3.6, and 4.8 mg/cm^2^ NPs were applied to EpiSkin RHE, which were 1, 6, 9, and 12 times the recommended dose, respectively.

Specifically, 7.5, 45, 67.5, and 90 mg/mL NP acetone suspension were configured at first and ultrasonically treated for 30 min, and then 20 μL NP acetone suspension was applied to the dermal surface. After acetone was volatilized, only NPs were retained in the epidermis, in concentrations of 0.4, 2.4, 3.6, and 4.8 mg/cm^2^. In the co-exposure group, the acetone suspension of ZnO NPs and TiO_2_ NPs needed to be mixed at the beginning, and then 40 μL mixed NP acetone suspension was applied on the EpiSkin surface to wait for acetone to volatilize.

After 24 h of NP treatment, the skin was washed with PBS three times to remove residual NPs, and thiazolyl blue (MTT) (HY-15924, MedChemExpress, Monmouth Junction, NJ, USA) was used to detect the viability of basal cells from EpiSkin after another 42 h incubation according to the protocol. As for the percutaneous penetration of NPs, the bottom medium was collected after 24 h exposure to NPs, and the epidermis layers and the collagen layers from the EpiSkin model were collected after another 42 h of incubation. After specific treatment of dilution and digestion, zinc (Zn) and titanium (Ti) elements in the tissues were measured by AAS (240FS AA (flame), Agilent Technologies, Palo Alto, CA, USA) and ICP-MS (7800, Agilent Technologies, Palo Alto, CA, USA) respectively.

### 2.12. Statistical Analysis

Data were presented as mean ± S.E.M from at least three independent experiments. Statistical differences between groups were compared by one-way analysis of variance (ANOVA). *p* < 0.05 was considered a significant difference. All graphics were prepared using GraphPad Softwar (Prism 6 e, La Jolla, CA, USA).

## 3. Results

### 3.1. Characterization of ZnO NPs and TiO_2_ NPs

ZnO NPs and TiO_2_ NPs are used in many fields of our daily life, and their adverse health responses are closely related to their physicochemical characteristics [38]. The morphology, crystal structure, and particle size of commercially available ZnO NPs and TiO_2_ NPs were characterized by cryogenic transmission electron microscopy (Cryo-TEM) and selected area electron diffraction (SAED). As shown in Figure 1, the presence of ZnO NPs in a spherical shape showed an average particle size of 33.21 nm (Figure 1A–C). TiO_2_ NPs with anatase crystal structure also had a spherical-like shape and an average diameter of 30.76 nm (Figure 1D–F). In order to better observe the cellular uptake of ZnO NPs in the subsequent experiments, FITC-labeled ZnO NPs were synthesized. It can be seen that the morphology of FITC-ZnO NPs did not change significantly after labeling, and their average diameter was 35.29 nm (Figure 1G–I).

The mean hydrodynamic diameters of ZnO NPs at 10, 20, and 100 μg/mL and TiO_2_ NPs at 10 and 100 μg/mL in DMEM with 10% FBS, determined by dynamic light scattering (DLS), were 74.55 ± 6.94 nm, 117.30 ± 5.44 nm, and 362.60 ± 14.73 nm and 302.08 ± 5.92 nm and 292.52 ± 0.61 nm, respectively. The hydrodynamic diameters of 10, 20, and 30 μg/mL FITC-ZnO NPs used in the following study were 210.20 ± 30.91 nm, 312.20 ± 32.27 nm, and 344.90 ± 13.80 nm (Table 1). The results demonstrated that these NPs had a certain degree of aggregation, and the agglomeration of ZnO NPs was concentration-dependent. In contrast, TiO_2_ NPs had no apparent concentration dependence in the culture medium. Compared with unlabeled ZnO NPs, the agglomerations of labeled FITC-ZnO NPs increased slightly. In addition, it could be seen that both ZnO and TiO_2_ NPs have zeta potentials between −15 and −10 mV (Table 1), possibly leading to moderate physical stability. For this reason, the NP suspensions used in our experiment were all freshly configured and prepared by powerful ultrasonication to prevent aggregation.

### 3.2. Co-Exposure with TiO_2_ NPs Reduced the Cytotoxicity Induced by ZnO NPs Alone

We firstly examined the cell viability of HaCaT keratinocytes exposed for 24 h to ZnO NPs and TiO_2_ NPs, respectively. The results showed that exposure to TiO_2_ NPs did not cause obvious cytotoxicity in HaCaT cells even when the concentration reached up to 300 μg/mL (Figure 2A). However, the cell viability was significantly inhibited upon ZnO NP exposure at 20 μg/mL, and it was observed that the HaCaT keratinocyte viability continued to decline with the increase in ZnO NP concentration, indicating that ZnO NPs had caused dose-dependent toxicity to HaCaT keratinocytes (Figure 2B).

ZnO NPs and TiO_2_ NPs are usually applied together in the formulation of sunscreens [39,40]. Therefore, we further explored the cytotoxicity produced by co-exposure of those two NPs with different proportions. We selected 20 μg/mL ZnO NPs, which was the toxic dose that inhibited the proliferation of keratinocytes, to co-incubate cells with different concentrations of TiO_2_ NPs for 24 h. Interestingly, TiO_2_ NPs offered some protective effects for the cytotoxicity induced by ZnO NPs. With the increase in TiO_2_ NP concentration from 100 μg/mL to 300 μg/mL, cell viability that was decreased by 20 μg/mL ZnO NPs gradually recovered to the level of the control group (Figure 2C). Further co-incubation of different concentrations of ZnO NPs (20, 30, 50, 100 μg/mL) with 100 μg/mL TiO_2_ NPs also showed a considerable decrease in the toxicity to HaCaT keratinocytes (Figure 2D). Meanwhile, a clear time-dependent response curve was obtained in both ZnO NP-treated and NP mixture-treated groups (Figure 2E). In addition, confocal microscope images obtained by calcein/PI (dead/living cells) dual staining intuitively supported the above conclusion that the cytotoxicity of HaCaT cells under co-exposure of ZnO NPs and TiO_2_ NPs was decreased compared with that of ZnO NP exposure alone (Figure 2F,G).

### 3.3. Co-Exposure with TiO_2_ NPs Reduced the DNA Damage Induced by ZnO NPs Alone

After revealing the cytotoxicity, we selected the comet assay, also known as the single-cell gel electrophoresis (SCGE) assay, to measure the strand breaks in the DNA of HaCaT cells. Results showed that ZnO NPs with different concentrations at 20, 50, and 100 µg/mL showed dose-dependent damage effects on the integrity of DNA where the percentage of DNA in tail (% of DNA in tail, TD%) increased with the increase in ZnO NPs (Figure 3A,C,E). In addition, TiO_2_ NPs demonstrated a potential to mitigate DNA damage caused by ZnO NPs, as evidenced by decreased TD% of 8.98%, 48.34%, and 55.66% compared to the original TD% of 34.25%, 77.84%, and 81.07% in the co-exposure system, respectively (Figure 3B,D,F). The above results indicated that TiO_2_ NPs displayed an antagonistic effect on DNA damage produced by ZnO NPs, which was consistent with the variation trend of cytotoxicity in the co-exposure system.

### 3.4. TiO_2_ NPs Reduced the Intracellular Content of Both ZnO NPs and Zn^2+^ Ions

Studies have revealed that at least 5~6 h of exposure was always required to reach the cellular absorption peak of ZnO NPs [41]. Therefore, to explore the concrete mechanism behind the antagonistic toxicity effects, we selected 20 μg/mL ZnO NPs and 100 μg/mL TiO_2_ NPs with an incubation time of 6 h in the following study. Firstly, flame atomic absorption spectroscopy (AAS) was used to detect the total zinc (Zn) content in HaCaT cells. Results showed that the Zn content in cells tended to increase following a gradual increase in the concentration of ZnO NPs (Figure 4A), indicating that the change in Zn content in HaCaT cells was concentration-dependent, which also indirectly confirmed the feasibility of AAS detection system in NP exposure study. After co-exposure with 100 μg/mL TiO_2_ NPs, the total amount of Zn element in 20 μg/mL ZnO NP-treated HaCaT cells decreased significantly (Figure 4B).

Given that the intracellular Zn element in our system was mainly contributed by ZnO NPs and Zn^2+^ ions, we next focused on the respective contributions of these two parts. First, we focused on detecting the change in ZnO NPs by using the silane coupling agent ATPS to prepare densely conjugated ZnO NPs with fluorescent substance FITC so that the change in particles could be measured by the change in fluorescence. Chemical structure analysis using a Fourier transform infrared spectrometer (FT-IR) showed that FITC-ZnO NPs had three prominent peaks at 3430 cm^−1^, 2972 cm^−1^, and 2925 cm^−1^, which appeared at the same position in the spectrum of FITC and correspond to its stretching vibrations of -OH, -CH_3_-, and -CH_2_- respectively, demonstrating that FITC had been successfully introduced to the surface of ZnO NPs (Figure 4C). In addition, the emission spectra of FITC-ZnO NPs confirmed this correct loading of FITC on ZnO NPs (Figure 4D); the emission peak of FITC-ZnO NPs at 520 nm was consistent with the emission intensity of FITC. Under confocal microscopy, FITC-ZnO NPs mainly showed green fluorescence at the excitation wavelength of 488 nm (Figure 4E). Meanwhile, FITC-ZnO NPs showed no difference in cytotoxicity compared to ZnO NPs without loading of FITC (Figure 4F).

Afterward, flow cytometry (FCM) was used to detect the mean fluorescence intensity (MFI) of HaCaT cells after exposure to FITC-ZnO NPs. Similar to the observations about the Zn element variation assessment, MFI showed a clear increase trend with FITC-ZnO NP exposure following the increase in particle concentration (Figure 4G). Yet, it was significantly reduced after co-exposure with TiO_2_ NPs (Figure 4H). Meanwhile, Zn^2+^ ions labeled with FluoZin-3 in HaCaT cells also showed an obvious downward trend during the co-exposure with TiO_2_ NPs (Figure 4I). Taken together, these data suggested that TiO_2_ NPs reduced the content of intracellular Zn element by limiting the entry of ZnO NPs into HaCaT cells and decreasing the amount of intracellular Zn^2+^ ions.

### 3.5. TiO_2_ NPs Increased the Particle Aggregation, Which Decreased the Cellular Uptake of ZnO NPs

Then, we decided to reveal the mechanism for uptake restriction upon the addition of TiO_2_ NPs to ZnO NPs. We observed the physical interaction between these NPs. As shown in Figure 5A–C, scanning transmission electron microscopy (STEM) images intuitively showed that, after co-exposure, ZnO NPs and TiO_2_ NPs (TiO_2_ NPs:ZnO NPs = 2:1) replaced the isolated state of their respective NPs with an apparent agglomeration phenomenon. Meanwhile, energy-dispersive X-ray spectroscopy (EDS) identified the elemental composition present in each STEM image (Figure 5D–F). Afterward, through analysis of hydrodynamic diameter identification, the association between the large size of NPs and physical agglomeration was further proven; the hydrodynamic diameter of the particles increased rapidly after the mixture of ZnO NPs and TiO_2_ NPs (Table 2). It has been reported that NPs undergoing agglomeration from original size to a large diameter could weaken both transmembrane and cellular uptake of the NPs [34,42]. The results presented herein suggest that the absorption of ZnO NPs was inhibited due to the physical aggregation caused by co-exposure with TiO_2_ NPs.

### 3.6. TiO_2_ NPs Restricted the Cellular Uptake of Non-Aggregated ZnO NPs by Competing for Caveolae-Mediated Endocytosis

Then, the attention shifted to another state of ZnO NPs, non-aggregated ZnO NPs. As reported, most NPs are taken up by cells through classical pathways: macropinocytosis, clathrin-mediated endocytosis, and caveolae-mediated endocytosis [34,43,44], while the uptake pathways of ZnO NPs and TiO_2_ NPs in HaCaT cells have not yet been elucidated. Therefore, we employed three inhibitors, namely chlorpromazine hydrochloride (CPZ), 5-(N-ethyl-N-isopropyl) amiloride (EIPA), and methyl-β-cyclodextrin (Mβ-CD). CPZ can specifically inhibit clathrin-mediated endocytosis, EIPA is an inhibitor of micropinocytosis, and Mβ-CD is a specific caveolin-mediated endocytosis inhibitor. Furthermore, we determined that the appropriate doses of these three inhibitors were CPZ ≤ 25 μM, EIPA ≤ 25 μM, and Mβ-CD ≤ 2.5 mM (Figure 6A–C). Afterward, the HaCaT cells were pretreated with these inhibitors for 1 h and then exposed to NPs for 6 h. Then, FCM was used to determine the concentration of intracellular FITC-ZnO NPs. The results demonstrated that FITC-ZnO NPs in all three inhibitor-treated groups underwent a certain extent of decrease. Still, only the Mβ-CD-treated group showed a statistically significant difference, indicating that caveolae-mediated endocytosis was the main pathway by which ZnO NPs enter HaCaT cells (Figure 6D). Moreover, the reduction in intracellular FITC-ZnO NPs caused by TiO_2_ NPs did not further increase the inhibitory effect of Mβ-CD, implying that there was no overlapping inhibitory effect between TiO_2_ NPs and Mβ-CD (Figure 6E). Subsequently, the cellular uptake of TiO_2_ NPs was determined by ICP-MS. It was found that the cellular uptake of TiO_2_ NPs was also concentration-dependent, and Mβ-CD significantly restricted the cellular uptake of TiO_2_ NPs (Figure 6F). It is noteworthy that the titanium content observed was less in the case of adding ZnO NPs than in the case in which ZnO NPs were absent (Figure 6F). Likewise, the reduction in intracellular titanium caused by ZnO NPs did not further increase the inhibitory effect of Mβ-CD on TiO_2_ NPs. In short, we conclude that both ZnO NPs and TiO_2_ NPs enter HaCaT cells through caveolae-mediated endocytosis, and these two types of NPs would compete with each other for absorption by HaCaT cells. So, TiO_2_ NPs restricted the cellular uptake of non-aggregated ZnO NPs by competing for caveolae-mediated endocytosis.

### 3.7. TiO_2_ NPs Decreased the Dissociation of ZnO NPs, thus Reducing the Content of the Main Toxic Contributor Zn^2+^

After clarifying the effects of TiO_2_ NPs on both aggregated and non-aggregated ZnO NPs, we continued to concentrate on another possible state of ZnO NPs in the co-exposure system, to wit, Zn^2+^. As mentioned above, we have proved that intracellular Zn^2+^ decreased in response to co-exposure with TiO_2_ NPs (Figure 4I), and now we need to elaborate on the deeper mechanisms. Generally speaking, Zn^2+^ mainly comes from the dissociation of Zn^2+^ from ZnO NPs, and the concentration of Zn^2+^ was always closely linked to the hydrolysis of ZnO NPs. Therefore, the detection of dissociated Zn^2+^ has gained much attention in the following study. As shown in Figure 7A, the concentration of dissociated Zn^2+^ in the NP suspension showed a dose-dependent relationship with the increase in ZnO NPs; however, the dissociation of ZnO NPs was restrained to a certain degree by TiO_2_ NPs (100 μg/mL) at both doses of 20 and 100 μg/mL. Furthermore, we showed that this inhibitory effect of TiO_2_ NPs on the dissociation of ZnO NPs would be strengthened over the accumulation of contact time (Figure 7B). Therefore, we deemed that the intracellular Zn^2+^ was limited by the decreased dissociation of ZnO NPs with the interaction of TiO_2_ NPs.

Thus far, we have demonstrated the mechanism underlying the restriction of the uptake of ZnO NPs into HaCaT cells and the decrease in the content of intracellular Zn^2+^ ions. However, it was still difficult to draw a conclusion regarding the respective contribution to joint toxicity of ZnO NPs and Zn^2+^. Therefore, Zn^2+^ chelating agent ethylene diamine tetraacetic acid (CaEDTA) was used for further study. The results showed that exposure to ZnO NPs significantly increased the concentration of Zn^2+^ in HaCaT cells, and CaEDTA could chelate intracellular Zn^2+^ to the basal level (Figure 7C). Meanwhile, chelating Zn^2+^ ions with CaEDTA completely rescued the cell viability reduced by ZnO NPs, even at high concentrations of 30 or 50 μg/mL (Figure 7D). These results suggested that the restriction of Zn^2+^ by TiO_2_ NPs might be the critical element for the reduction in cytotoxicity in the co-exposure system. In addition, after inhibition of caveolae-mediated endocytosis, there was an intensive increase in cytotoxicity rather than a recovery of the cell viability decreased by ZnO NPs (Figure 7E) since the intracellular Zn^2+^ was elevated under this condition (Figure 7F). Accordingly, we speculated that when the uptake of ZnO NPs was inhibited, the confinement of ZnO NPs in the extracellular region might increase the concentration of Zn^2+^ that flooded into cells and induced cell death, which also indirectly confirmed the main contribution of Zn^2+^ to cytotoxicity.

Taken together, our results preliminarily revealed the mechanism of joint toxicity induced by ZnO NPs and TiO_2_ NPs in HaCaT cells. On the one hand, TiO_2_ NPs increase the agglomeration of particles and thus confine the absorption of aggregated ZnO NPs. On the other hand, TiO_2_ NPs compete for caveolae-mediated endocytosis, thus limiting the cell uptake of non-aggregated ZnO NPs. Meanwhile, TiO_2_ NPs decrease the dissociation of ZnO NPs to Zn^2+^, and the shrinkage of intracellular Zn^2+^ ultimately makes TiO_2_ NPs perform an antagonistic effect on the cytotoxicity caused by ZnO NPs.

### 3.8. TiO_2_ NP Co-Exposure Reduced the Dermal Toxicity Induced by ZnO NPs Alone on the In Vitro Reconstructed Human Epidermis (RHE) EpiSkin

Our results showed that ZnO NPs caused no significant dermal damage at the recommended concentration of 0.4 mg/cm^2^, but decreased the activity of basal cells to 70% and 40% when the dosage increased to 9 or 12 times the recommendation; that is, application of ZnO NPs at 3.6 or 4.8 mg/cm^2^ was toxic to the in vitro epidermis (Figure 8B). In our system, the application of TiO_2_ NPs to skin models did not cause any dermal mortality even at the usage of 12 times (4.8 mg/cm^2^) the recommendation (Figure 8C). Consistent with the cytotoxicity, co-exposure of TiO_2_ NPs at 4.8 mg/cm^2^ significantly reduced the dermal toxicity induced by ZnO NP exposure alone (Figure 8D).

As is well known, enough evidence has been given for the dermal toxicity of NPs to be associated with percutaneous penetration [45,46]. Therefore, after NP application, we recycled and digested the model tissues of epidermal layers, collagen layers, and bottom culture medium and tested the permeability of Zn and Ti elements in skin tissues by AAS and ICP-MS, respectively. On the whole (Figure 8E–J), massive deposits of NPs were growing into the epidermal layer, while the collagen layer and bottom medium only held a tiny spot in NP amounts. In the epidermal layers (Figure 8E,F), the skin infiltration deposition of TiO_2_ NPs and ZnO NPs was dose-dependent, and the epidermal elements of Ti and Zn increased significantly with the increased application of TiO_2_ NPs and ZnO NPs. Consistent with the toxicity reduction, the epidermal permeability of ZnO NPs was significantly decreased by co-exposure with TiO_2_ NPs. In the collagen layers (Figure 8G,H), the deposition of ZnO NPs also showed a dose-dependent manner, but there was no statistical difference in the TiO_2_ NP-treated group. After co-exposure, the deposition of Zn in the collagen layer also decreased significantly. In the bottom medium (Figure 8I,J), we only detected the presence of a small amount of Zn element, and no absorption of the TiO_2_ NPs in the bottom medium was observed. The variation of Zn element in the co-exposure group was also in accordance with the above decreasing tendency.

These results revealed that ZnO NPs could penetrate epidermal layers and collagen layers into the deep bottom medium and cause apparent dermal toxicity. In the co-exposure system, TiO_2_ NPs restrict the penetration of ZnO NPs and thus reduce the dermal toxicity induced by ZnO NPs.

## 4. Discussion

Over the past two decades, nanomaterials have been a rising star in modern materials chemistry and engineering that has contributed immensely to advancing the frontiers of technology. With the wide application of NPs in daily life products and industrial manufacturing environments, the exposure risk increases, which has drawn additional attention to the safety of NPs for humans. Exposure to NPs can occur through ingestion, inhalation, or dermal absorption, thus causing different degrees of potential health risks to human bodies [47,48,49]. As the largest organ of humans, the skin is the first line of defense protecting our bodies from external perturbations [50]. Therefore, the health risks to human skin arising from NPs raised particular concern among people. Being the main components of sunscreens, ZnO NPs and TiO_2_ NPs are often used in different brands of sunscreen products with different proportions. However, the joint skin effect and molecular mechanism between ZnO NPs and TiO_2_ NPs were still unclear. Therefore, this work aimed to explore the dermal toxicity caused by co-exposure of ZnO NPs and TiO_2_ NPs at the cellular level as well as the EpiSkin model level, in the hope of revealing the underlying mechanism and, on this basis, suggesting a safer sunscreen ingredient formula.

Although some studies on TiO_2_ NPs indicated their role in ROS generation [51,52], inflammation, or DNA damage [49] in skin cells, such as HaCaT cells and Balb/c mice skin cells [52,53], our study showed that TiO_2_ NPs within the dose range of experiment caused no apparent cytotoxicity or DNA damage to keratinocytes. A study involving the proliferation of HaCaT cells also observed no significant reduction in cell viability at 24 h upon treatment of ultrafine TiO_2_ NPs (UF-TiO_2_), but rather the promotion of the cell viability in the range of low dosage (0.1~10 μg/mL) [54]. Generally, TiO_2_ NPs tend to show lower toxicity than other metal oxide NPs [55]. However, the toxic effect of ZnO NPs on skin cells seems to be recognized by most researchers [56,57]. Meanwhile, plentiful studies revealed their cytotoxicity and DNA damage effects on HaCaT cells [58,59,60], which was consistent with our results.

In recent years, the health hazards caused by co-exposure to different NPs have gradually attracted attention from the scientific community [61]. A summary of the results published so far about the combined toxicity of ZnO NPs and TiO_2_ NPs reveals two different conclusions. The first is the synergistic toxicity interaction between ZnO NPs and TiO_2_ NPs. For instance, Opeoluwa M. Ogunsuyi et al. [28] found that both NPs and their mixture apparently altered sperm motility, reduced sperm numbers, and increased abnormalities, while their mixture induced more sperm abnormalities than either TiO_2_ NPs or ZnO NPs alone. In addition, in human Jurkat cells, TiO_2_ NPs also increased the cytotoxicity of ZnO NPs but reduced the phosphorylation of some signaling proteins, thus suggesting that the synergistic toxic effect may be related to intracellular activation routes [29]. However, some studies have demonstrated the antagonistic toxicity between ZnO NPs and TiO_2_ NPs, including our conclusions. Yu, Wu, Liu, Chen, Zhu, and Lu [27] found that co-existing TiO_2_ NPs displayed a dose-dependent mitigation effect on the cytotoxicity of ZnO NPs to a bacterium. Another study focusing on 30 groups of binary mixtures, including different proportions of ZnO NPs, NiO NPs, CuO NPs, TiO_2_ NPs, and Fe_2_O_3_ NPs, found that additive action accounted for 67%, followed by synergistic effect (16.5%) and antagonistic activity (16.5%), and the combination of ZnO NPs and TiO_2_ NPs showed antagonistic toxicity to the chlorophyll content of algae [62]. Subsequently, a hint associated with this antagonistic effect pointed at the mechanism of substantial adsorption of Zn^2+^ by TiO_2_ NPs [63]. However, another potential theory of aggregate association between ZnO NPs and TiO_2_ NPs has also explained the recuperative acetylcholinesterase (AchE) activity in the brain of juvenile fish [64]. However, studies on the combined effects of ZnO NPs and TiO_2_ NPs on skin-related cells and skin models were very limited in previous studies.

In this study, we evaluated the toxic effects caused by the co-exposure of ZnO NPs and TiO_2_ NPs. We noticed that the mixture of NPs was not only a phenomenon of environmental pollution in water or soil, and the combined application of different NPs in personal care products should be given sufficient attention. In this regard, our results contributed that TiO_2_ NPs could play an antagonistic effect on the cytotoxicity and DNA damage induced by ZnO NPs. In fact, early in 2014, a study simply explored the joint effect of ZnO NPs and TiO_2_ NPs on the epidermal model KeraSkin. However, it only found that 45 min exposure to 25% ZnO NPs resulted in slight damage to the skin model (basal cell viability decreased to 80%), and co-exposure of ZnO NPs and TiO_2_ NPs caused no irritation or corrosion to the epidermis [65]. Higher dose exposure and mechanism studies are lacking.

As reported, the level of cellular uptake of NPs was strongly dependent on their agglomeration state [66]. This suggested that agglomerated NPs, which have in their agglomerated state a bigger size than individually dispersed NPs, should be internalized less than individual NPs [67,68]. Abdelmonem et al. [69] found that 3T3 fibroblasts and HeLa cells have lower uptake for the larger-sized agglomerates than the smaller dispersed NPs, which was consistent with our discovery for agglomerated ZnO NPs in keratinocytes. Therefore, we speculated that the strong agglomeration of ZnO NPs with TiO_2_ NPs increased the hydrodynamic diameter of particles in the transmembrane process, which made the actual size larger than commercially labeled so that the ZnO NPs in this state were restricted from entering cells. Generally, NPs with physical diameters of approximately equal to or less than one-half of the light wavelength would reflect and scatter the most significant amount of light and accomplish the best UV shielding effect. To maximize the protective efficiency of ZnO NPs against UVA (320–400 nm), the primary particle sizes of ZnO NPs in commercial sunscreens are mostly fabricated between 10 and 200 nm, and mainly the grades with larger particles are used [8,11]. On the one hand, the agglomeration effect of TiO_2_ NPs on ZnO NPs limits the cellular uptake of harmful components; meanwhile, it also ensures the UVA shielding effect produced by ZnO NPs. However, the problem is that intensive agglomeration may also influence the optical properties of TiO_2_ NPs. Given that the UVB absorption of TiO_2_ NPs is size-dependent, the TiO_2_ NPs with large diameters are usually not the optimal choice for sunscreen products [70]. Therefore, determining how to balance the relationship between NP agglomeration and UV protection effects still needs further study.

The classical pathways of NPs internalization mainly include macropinocytosis, clathrin-mediated endocytosis, and caveolae-mediated endocytosis [34,43]. Tian Xia et al. [71] found that caveolae proteins dominated the localization of undissolved ZnO NPs in human bronchial epithelial cells (BEAS-2B). Moreover, another systematic study [42] in HaCaT cells showed that larger ZnO NPs (70 nm) entered HaCaT cells through the caveolae-mediated pathway while smaller ZnO NPs (20 nm) were internalized into cells mainly by clathrin-mediated endocytosis, which suggested that the uptake of ZnO NPs seemed to be size-dependent. After well dispersing, ZnO NPs showed an average particle size of 33.21 nm in our study, and we confirmed their main internalization route as the caveolae-mediated pathway. Obviously, our conclusion was consistent with the previous research. In addition, Tian Xia et al. [71] also observed that TiO_2_ NPs were also taken up intact into caveolin-1 positive endosomal compartments in BEAS-2B cells. Likewise, it was found that in human glial cells U373, TiO_2_ NPs increased the expression of CAV-1, a marker protein for caveolae [72]. Just as we found, TiO_2_ NPs can be absorbed by HaCaT cells through caveolae-mediated endocytosis.

Moreover, further investigation suggested that TiO_2_ NPs played a competitive inhibitory effect on the uptake of non-aggregated ZnO NPs. Similar phenomena include competitive inhibition of liver uptake between Fe_3_O_4_ NPs and CdCl_2_ by sharing divalent metal transporters [73] and the decreased bioavailability of hematite (HemNPs) to Ag NPs in Ochromonas danica [74]. In addition, Bin Huang et al. [75] also made a similar conclusion in the study of polyacrylate-coated TiO_2_ NPs and α-Fe_2_O_3_ NPs in the protozoan Tetrahymena thermophila, discovering that the significant uptake competition between these two types of NPs limited their bioaccumulation of each other. Accordingly, for the first time, we elucidated the competitive effect of TiO_2_ NPs and ZnO NPs on caveolae-mediated endocytosis from the perspective of cell uptake and speculated that this might be an embodiment of the interaction of NP co-exposure with skin cells.

Thus far, it has still been difficult to draw a conclusion whether the cytotoxicity of ZnO NPs is dominated by NPs themselves or contributed by dissolved Zn^2+^. Doctor Amy Holmes [76] once found that ZnO NPs appeared to be more toxic than ZnSO_4_ at equivalent zinc concentrations to HaCaT keratinocytes, indicating that the toxicity of ZnO NPs was related to labile Zn^2+^ but also total ZnO NPs. However, Philip J. Moos et al. [77] reported that direct particle–cell contact was required for RKO cell cytotoxicity and observed that ZnO particulate matter caused mitochondrial dysfunction, while these toxic effects were independent of the amount of soluble Zn^2+^. Nevertheless, most studies tended to support the standpoint that Zn^2+^ plays a major role in toxicity produced by ZnO NPs. It is generally admitted that in a non-phagocytic system, the toxicity of insoluble NPs is related to their crystal structure and atomic bonding properties, while the toxicity of soluble NPs is mainly determined by their solubility [78]. For example, in human umbilical vein endothelial cells (HUVECs), ZnSO_4_ induced cytotoxic effects similar to those of ZnO NPs, which indicated the major toxic effect of Zn^2+^ [17]. In human bronchial epithelial cells, the proinflammatory response and mitochondrial damage were directly related to the released Zn^2+^ from ZnO NPs [71]. Consistent with our conclusion, we found that chelating Zn^2+^ with CaEDTA completely restored the cell viability decreased by ZnO NPs, indicating that Zn^2+^ was the main contributor to HaCaT cytotoxicity.

As reported, the physicochemical properties and potential toxic effects of soluble NPs, such as ZnO NPs, Ag NPs, and Fe_2_O_3_ NPs, are likely to be influenced by other co-existing insoluble NPs [79]. With a diameter of 20–50 nm, TiO_2_ NPs can bind and accumulate metal ions onto their surface due to their relatively loose particles and the large specific area, which made TiO_2_ NPs widely used as environment-purifying catalysts [80]. Likewise, we also revealed the limiting effect of TiO_2_ NPs on the dissociation of ZnO NPs, which was very similar to a report in 2014 that found that the combined effects of ZnO NP dissolution and Zn^2+^ adsorption onto TiO_2_ NPs controlled the concentration of dissolved Zn^2+^ [79]. In addition, Tang et al. [81] also proposed that the presence of TiO_2_ NPs at low concentrations enhances the toxicity of Zn^2+^. Still, the toxicity of Zn^2+^ significantly decreased with increasing TiO_2_ NP concentration because of the substantial adsorption of Zn^2+^ by TiO_2_ NPs. Accordingly, we speculated that the Zn^2+^ reduction observed in our study might partly contribute to the effect of adsorption of dissociated Zn^2+^ by TiO_2_ NPs. Moreover, the control of TiO_2_ NPs on the main toxic substance Zn^2+^ finally made it play an antagonistic role in the co-exposure system.

Finally, we evaluated the joint skin effect induced by NP mixtures on the epidermal model EpiSkin. As in historical studies [18,19,20,65], no apparent dermal damage from TiO_2_ NPs was observed in our study, regardless of their concentration. However, the dermal toxicity increased significantly under high-dose exposure to ZnO NPs. Percutaneous penetration of native skin plays an essential role in dermal toxicity [45,46]. For example, when ZnO NPs in sunscreen were absorbed by healthy skin of people, increased Zn levels were detected in blood and urine [74]. In addition, topical application of ZnO NPs to newborn mice also revealed cuticle permeability to Zn element, showing the deeper penetration of ZnO NPs caused accumulation of Zn within hair follicles and thus induced the apoptosis of hair follicle stem cells (HFSCs) [82]. The uptake of ZnO NPs in THP-1 monocyte-derived human macrophages has also been reported to suggest that a small amount of Zn penetrates the skin of healthy people and enters the bloodstream, which could stimulate the immune response [83]. These results were consistent with our study, which showed that ZnO NPs did undergo percutaneous penetration at the dosage that induced dermal toxicity. The reduction in dermal toxicity after NP co-application was closely related to the decrease in NP penetration. On the surface of the EpiSkin model is the corneum, a tightly connected layer of dozens of keratinocytes, which is the first barrier for the entry of NPs to deeper skin. Based on the interaction effects between TiO_2_ NPs and ZnO NPs in HaCaT cells, we speculated that the uptake inhibition of TiO_2_ NPs on ZnO NPs in keratinocytes was also the main reason which caused the reduction in the transdermal permeability of ZnO NPs after co-exposure.

Last but not the least, we would like to have an additional discussion on the latest regulations and related precautions for ZnO and TiO_2_ used in cosmetics. At present, many countries have approved the addition of ZnO and TiO_2_ in cosmetics, and the latest documents shown in Table 3 demonstrate that the maximum dose of ZnO and TiO_2_ cannot exceed 25%. The regulations of China and the United States do not clearly distinguish between nano and non-nano physical UV filters. The European Commission has specified that sum of the highest concentration of ZnO and ZnO NP share and TiO_2_ and TiO_2_ NP share shall not exceed 25%, the aerodynamic diameter of powdered titanium dioxide shall be less than 10 μm, and ZnO and ZnO NPs should not be used in applications that may lead to exposure of the end-user’s lungs by inhalation.

Moreover, as found in our study, the stability of NPs is closely related to their toxic effects. The ideal sunscreen formulation should ensure an ample photoprotective effect against UVA and UVB with a minimal amount of filters while reducing toxicity to its lowest possible level. Therefore, an emulsifier becomes an essential part of a sunscreen formula as it plays an important role in emulsifying and dispersing the oil and active ingredients in the formula and provides special skin feeling and rheological properties. We researched dozens of common sunscreens containing physical UV filters and found that the commonly used emulsifiers are polyoxyethylene stearates, polyglycerol stearates, polysiloxanes, etc., of which PEG-9 polydimethylsiloxyethyl dimethicone is the most common. However, none of the products specify whether their physical sunscreen ingredients are nanoscale, perhaps to avoid consumer panic. In order to improve the stabilization of sunscreen NPs, relevant companies need to create more effective emulsifiers. Surfactant-free Pickering emulsions can be stabilized by the UV filter NPs for the manufacture of sunscreen products [6,88]. In addition, the skin effect of chronic exposure to NPs should also be given attention. Unfortunately, this paper did not focus on an in-depth study about this, and we hope it can be supplemented in the future.

## 5. Conclusions

In this study, we revealed the antagonistic joint skin toxicity of co-exposure to physical sunscreen ingredients TiO_2_ NPs and ZnO NPs and analyzed the inhibition effect of TiO_2_ NPs on the cellular uptake of aggregated ZnO NPs and non-aggregated ZnO NPs, proposing for the first time the competitive inhibition effect of these two NPs on caveolae-mediated endocytosis. Furthermore, we verified these antagonistic joint skin effects on the in vitro RHE model EpiSkin and further demonstrated that the ZnO NPs transdermal restriction induced by TiO_2_ NPs played an essential role in decreasing the dermal toxicity of the co-exposure system. Our work provides a new perspective for explaining combined toxicity caused by different NPs. More importantly, we contributed some suggestions on the proportion formula for the safer application of ZnO NPs and TiO_2_ NPs in sunscreens.

## Figures and Tables

**Figure 1 nanomaterials-12-02769-f001:**
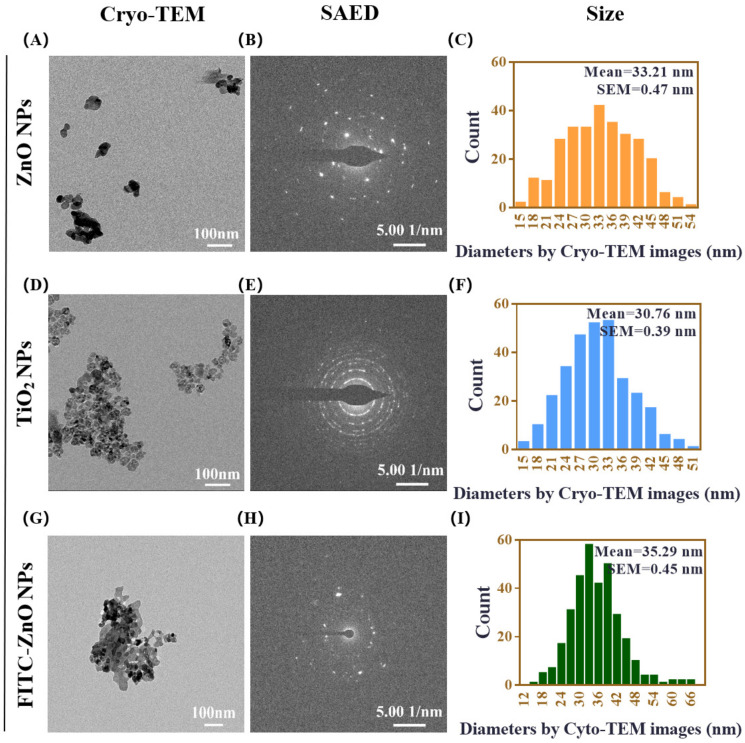
Characterization of ZnO NPs and TiO_2_ NPs. (**A**–**C**) The representative morphologies of ZnO NPs shown using cryogenic transmission electron microscopy (Cryo-TEM) (**A**), the crystal structure of ZnO NPs analyzed using selected area electron diffraction (SAED) (**B**), and size-distribution histograms of ZnO NPs obtained using Nano Measurer software (**C**). (**D**–**F**) The representative morphologies of TiO_2_ NPs shown using Cryo-TEM (**D**), the crystal structure of TiO_2_ NPs analyzed using SAED (**E**), and size-distribution histograms of TiO_2_ NPs obtained using Nano Measurer software (**F**). (**G**–**I**) The representative morphologies of FITC-ZnO NPs shown using cryogenic transmission electron microscopy (Cryo-TEM) (**G**), the crystal structure of FITC-ZnO NPs analyzed using selected area electron diffraction (SAED) (**H**), and size-distribution histograms of FITC-ZnO NPs obtained using Nano Measurer software (**I**). Cryo-TEM scale bar = 100 nm, SAED scale bar = 5.00 1/nm, counts include more than 300 particles in each case of size analysis.

**Figure 2 nanomaterials-12-02769-f002:**
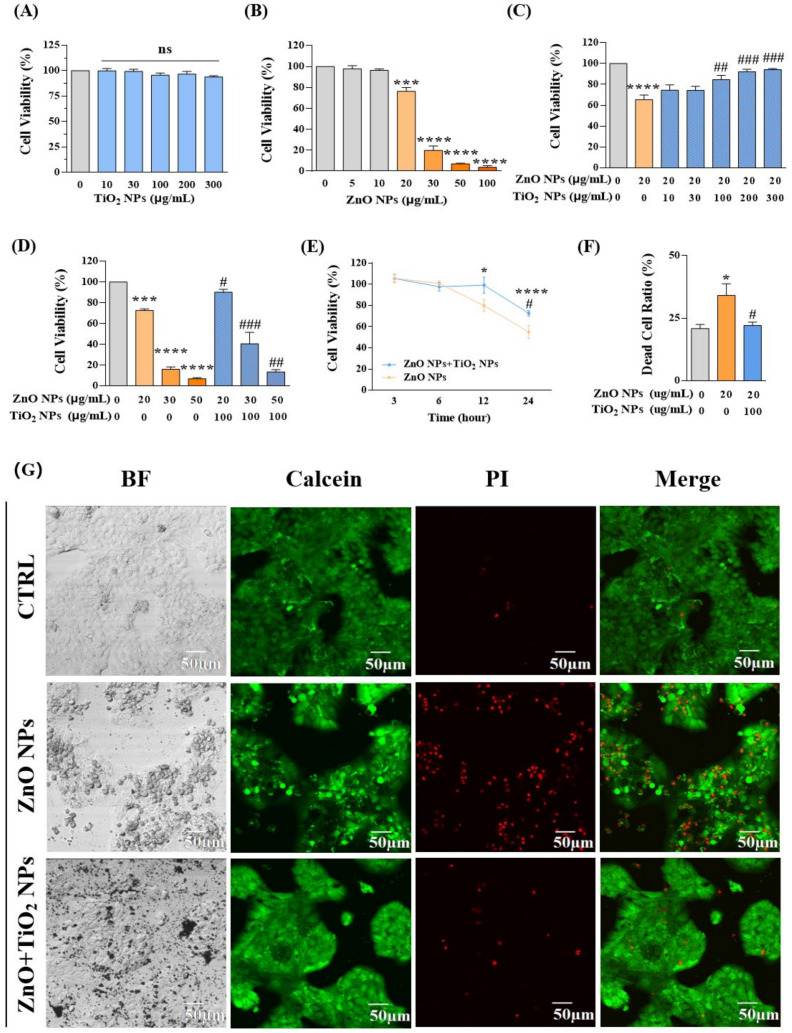
Co-exposure with TiO_2_ NPs reduced the cytotoxicity induced by ZnO NPs alone. (**A**,**B**) Cell viability was measured at different doses of (**A**) TiO_2_ NPs (0, 10, 30, 100, 200, and 300 μg/mL) and (**B**) ZnO NPs (0, 5, 10, 20, 30, 50, and 100 μg/mL). (**C**,**D**) Effects on cell viability caused by co-exposure of ZnO NPs and TiO_2_ NPs. (**E**) Time-dependent toxicity curves were obtained in both ZnO NP-treated and NP mixture-treated groups for 3, 6, 12, and 24 h. (**F**,**G**) Confocal microscope images (**G**) and statistical analysis (**F**) for calcein/PI (dead/living cells) dual staining. * *p* < 0.05, *** *p* < 0.001, **** *p* < 0.0001 compared to the control group. ^#^
*p* < 0.05, ^##^
*p* < 0.01, ^###^
*p* < 0.001 compared to ZnO NP-treated group.

**Figure 3 nanomaterials-12-02769-f003:**
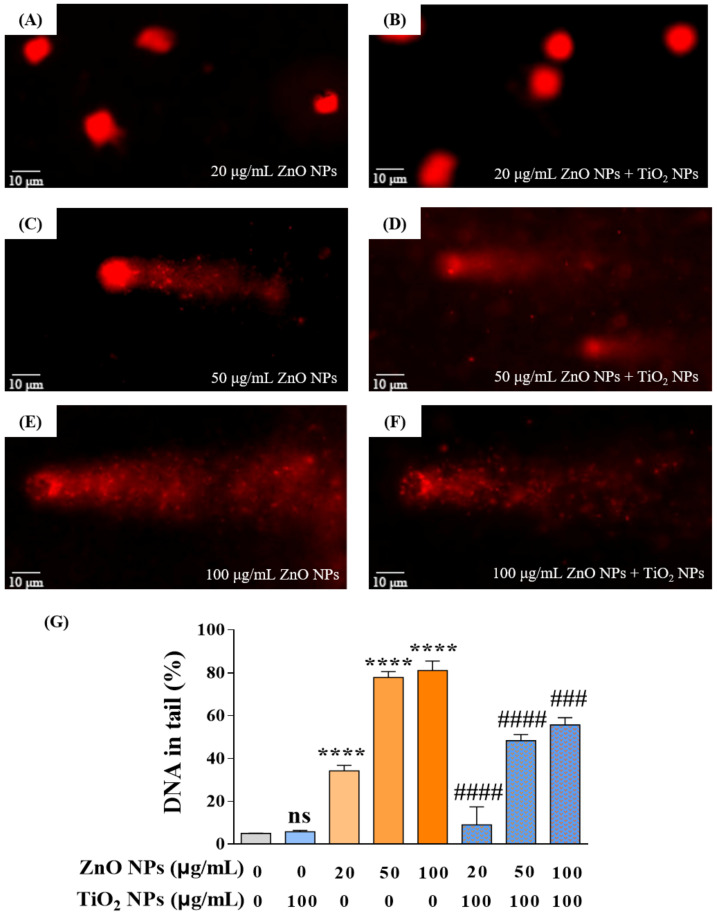
Co-exposure with TiO_2_ NPs reduced the DNA damage induced by ZnO NPs alone. (**A**,**C**,**E**) DNA damage of HaCaT cells induced by ZnO NPs at 20 (**A**), 50 (**C**), and 100 (**E**) μg/mL. (**B**,**D**,**F**) DNA damage of HaCaT cells induced by co-exposure of ZnO NPs (20 (**B**), 50 (**D**), and 100 (**F**) μg/mL) with 100 μg/mL TiO_2_ NPs. (**G**) Statistical analysis for DNA damage. **** *p* < 0.0001 compared to control group. ^###^
*p* < 0.001, ^####^
*p* < 0.0001 compared to ZnO NP-treated group.

**Figure 4 nanomaterials-12-02769-f004:**
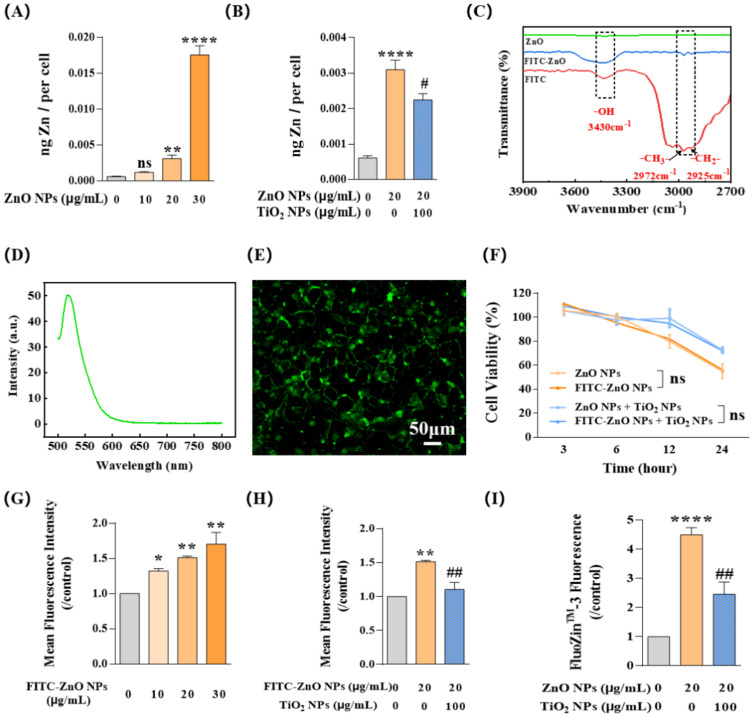
TiO_2_ NPs reduced the intracellular content of both ZnO NPs and Zn^2+^ ions. (**A**,**B**) Intracellular Zn content was measured by flame atomic absorption spectroscopy (AAS) at different doses of ZnO NPs (0, 10, 20, and 30 μg/mL) with (**B**) or without (**A**) TiO_2_ NPs. (**C**) Fourier transform infrared spectrometer (FT-IR) measurements proved that FITC had been successfully introduced to the surface of ZnO NPs. (**D**) Emission spectra of FITC-ZnO NPs confirmed that FITC was successfully loaded onto particles. (**E**) FITC-ZnO NPs showed green fluorescence at the excitation wavelength of 488 nm when incubated with HaCaT cells. (**F**) FITC-ZnO NPs showed no difference in cytotoxicity compared to ZnO NPs without loading of FITC. (**G**,**H**) The cellular uptake of FITC-ZnO NPs with (**H**) or without (**G**) TiO_2_ NPs was investigated by flow cytometry (FCM). (**I**) The intracellular Zn^2+^ ions indicated by FluoZin-3, a Zn^2+^ specific probe, were investigated by FCM. * *p* < 0.05, ** *p* < 0.01, **** *p* < 0.0001 compared to the control group. ^#^
*p* < 0.05, ^##^
*p* < 0.01 compared to ZnO NP-treated group.

**Figure 5 nanomaterials-12-02769-f005:**
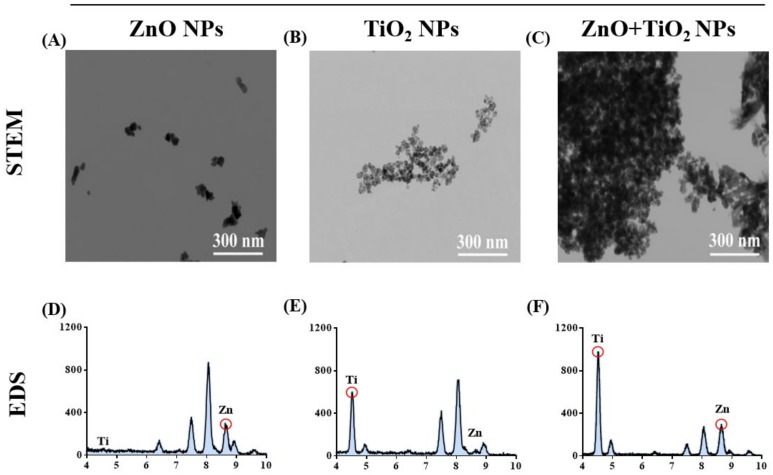
TiO_2_ NPs increased the particle aggregation, which decreased the cellular uptake of ZnO NPs. (**A**–**C**) The agglomeration state of ZnO NPs (**A**), TiO_2_ NPs (**B**), and their mixtures (**C**) was observed by scanning transmission electron microscopy (STEM). (**D**–**F**) The elemental composition present in each STEM image of ZnO NPs (**D**), TiO_2_ NPs (**E**), and their mixtures (**F**) was identified by energy-dispersive X-ray spectroscopy (EDS).

**Figure 6 nanomaterials-12-02769-f006:**
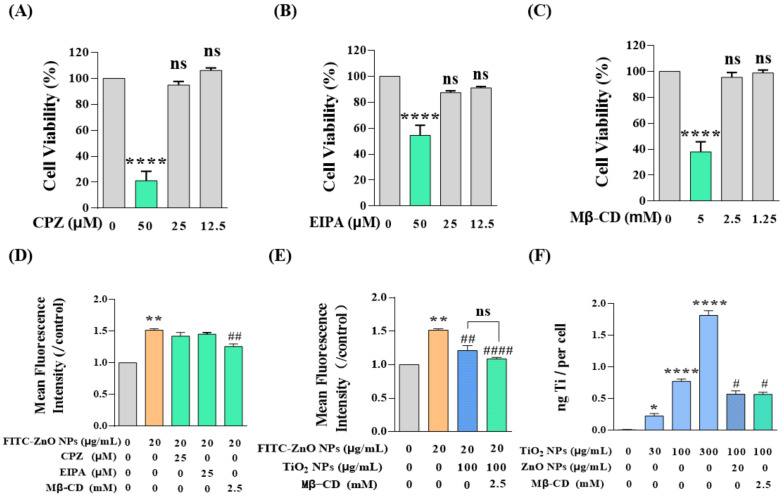
TiO_2_ NPs restricted the cellular uptake of non-aggregated ZnO NPs by competing for caveolae-mediated endocytosis. (**A**–**C**) The appropriate doses of inhibitors determined by CCK-8 assay were CPZ ≤ 25 μM (**A**), EIPA ≤ 25 μM (**B**), and Mβ-CD ≤ 2.5 mM (**C**). (**D**–**E**) Intracellular FITC-ZnO NPs were detected by FCM after treatment with three endocytosis inhibitors with (**E**) or without (**D**) co-exposure of TiO_2_ NPs. (**F**) Intracellular titanium contents on behalf of TiO_2_ NPs were measured by ICP-MS. * *p* < 0.05, ** *p* < 0.01, **** *p* < 0.0001 compared to the control group. ^#^
*p* < 0.05, ^##^
*p* < 0.01, ^####^
*p* < 0.0001 compared to the single ZnO NP-treated group or the single TiO_2_ NP-treated group.

**Figure 7 nanomaterials-12-02769-f007:**
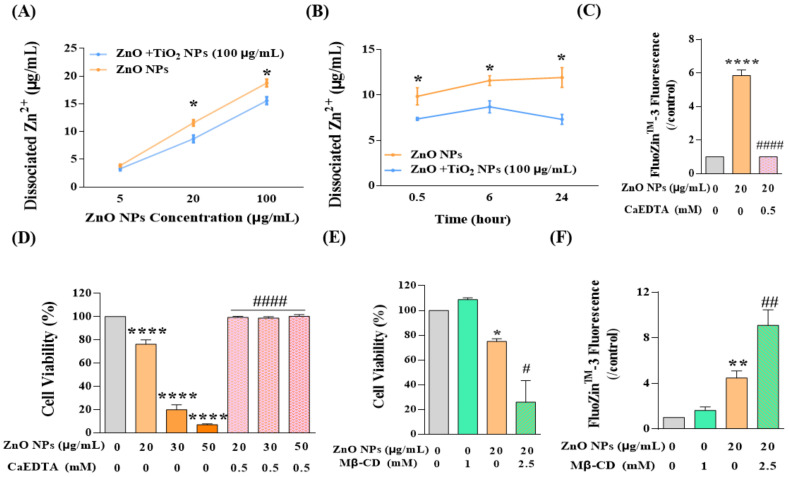
TiO_2_ NPs decreased the dissociation of ZnO NPs, thus reducing the content of the main toxic contributor Zn^2+^. (**A**,**B**) The dissociated Zn^2+^ of ZnO NPs at (**A**) different concentration (5, 20, 100 μg/mL) and (**B**) different time intervals (0.5, 6, 24 h) was detected by AAS. (**C**–**F**) Cytotoxicity and intracellular Zn^2+^ ions were detected after the treatment of ZnO NPs with or without CaEDTA (**C**,**D**) and Mβ-CD (**E**,**F**). * *p* < 0.05, ** *p* < 0.01, **** *p* < 0.0001 compared to the co-exposure group or the control group. ^#^
*p* < 0.05, ^##^
*p* < 0.01, ^####^
*p* < 0.0001 compared to ZnO NP-treated group.

**Figure 8 nanomaterials-12-02769-f008:**
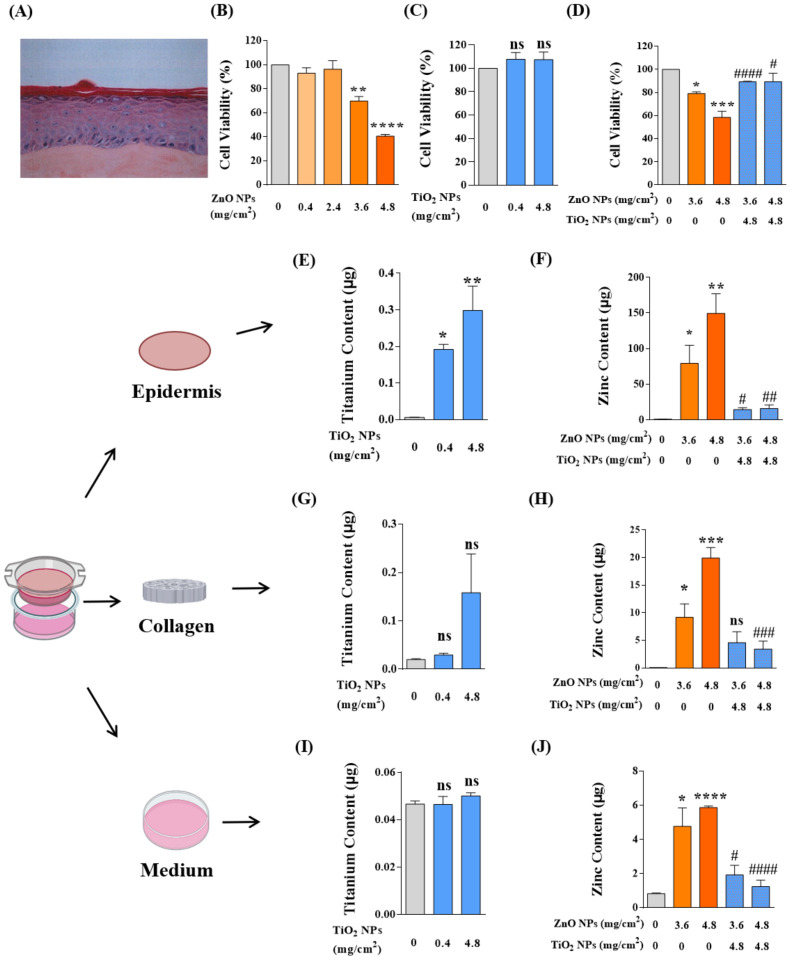
TiO_2_ NP co-exposure reduced the dermal toxicity induced by ZnO NPs alone on the in vitro reconstructed human epidermis (RHE) EpiSkin. (**A**) The hematoxylin–eosin staining (HES) of paraffin sections of EpiSkin model. (**B**,**C**) The viability of basal cells from EpiSkin epidermis was measured after treatment with ZnO NPs (**B**) at 0, 0.4, 2.4, 3.6, and 4.8 mg/cm^2^ or TiO_2_ NPs (**C**) at 0, 0.4, and 4.8 mg/cm^2^. (**D**) The dermal toxicity induced by co-exposure of ZnO NPs and TiO_2_ NPs was evaluated by MTT assay. (**E**,**F**) Percutaneous penetration of TiO_2_ NPs (**E**), ZnO NPs, and their mixtures (**F**) was measured by ICP-MS or AAS in epidermis layers. (**G**,**H**) Percutaneous penetration of TiO_2_ NPs (**G**), ZnO NPs, and their mixtures (**H**) was measured by ICP-MS or AAS in collagen layers. (**I**,**J**) Percutaneous penetration of TiO2 NPs (**I**), ZnO NPs, and their mixtures (**J**) was measured by ICP-MS or AAS in bottom medium. * *p* < 0.05, ** *p* < 0.01, *** *p* < 0.001, **** *p* < 0.0001 compared to the control group. ^#^
*p* < 0.05, ^##^
*p* < 0.01, ^###^
*p* < 0.001, ^####^
*p* < 0.0001 compared to the single ZnO NP-treated group.

**Table 1 nanomaterials-12-02769-t001:** The hydrodynamic diameter and zeta potential of ZnO and TiO_2_ NPs in DMEM high sugar medium with 10% FBS.

Nanoparticles	Concentration (µg/mL)	Zeta Potential (ZP, mV)	Hydrodynamic Size (HDS, nm)
ZnO NPs	100	−11.20 ± 1.50	362.60 ± 14.73
	20	−12.80 ± 0.21	117.30 ± 5.44
	10	−11.21 ± 0.19	74.55 ± 6.94
TiO_2_ NPs	100	−12.12 ± 0.16	292.52 ± 0.61
	10	−12.81 ± 0.33	302.08 ± 5.92
FITC-ZnO NPs	30	−15.48 ± 0.58	344.90 ± 13.80
	20	−13.90 ± 0.89	312.20 ± 32.27
	10	−14.23 ± 0.44	210.20 ± 30.91

**Table 2 nanomaterials-12-02769-t002:** The hydrodynamic size of ZnO, TiO_2_ NPs, and their mixtures in DMEM high sugar medium with 10% FBS.

Nanoparticles	Concentration (µg/mL)	Hydrodynamic Size (HDS, nm)
ZnO NPs	20	117.3 ± 5.44
TiO_2_ NPs	100	315.6 ± 3.50
ZnO + TiO_2_ NPs	20 + 100	1007 ± 58.20 **** ^####^

**** *p* < 0.0001 compared to ZnO NP-treated group. ^####^
*p* < 0.0001 compared to TiO_2_ NP-treated group.

**Table 3 nanomaterials-12-02769-t003:** A brief review of the safety and technical standards for cosmetics containing ZnO and TiO_2_.

Safety and Technical Standards for Cosmetics			Regulations
Countries	Time	Documents and Institutions	
China	2022	Safety and Technical Standards for Cosmetics [84] National Institutes for Food and Drug Control (NIFDC)	The maximum allowable concentration of TiO_2_ and ZnO used in sunscreen is 25%.
	2012	GB/T 27599 Titanium dioxide for cosmetic use [85] State Administration for Market Regulation	The TiO_2_ used for cosmetics is classified into two types: type I (no surface treatment) and type II (after surface treatment).
United States	2021	Over-the-Counter Monograph M020: Sunscreen Drug Products for Over-the-Counter Human Use [86] Food & Drug Administration (FDA)	The maximum allowable concentration of TiO_2_ and ZnO used in sunscreen is 25%.
Europe	2022	Regulation (EC) No 1223/2009 of the European Parliament and of the Council of 30 November 2009 on cosmetic products (recast) (OJ L 342, 22.12.2009, p. 59) [87] European Commission	The same as for TiO_2_, the maximum concentration of ZnO (nano) in ready-for-use preparation is 25%. Both of them should not be used in applications that may lead to exposure of the end-user’s lungs by inhalation.

## Data Availability

The data presented in this study are available on request from the corresponding author.

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
