# Peer review of "Antagonistic Skin Toxicity of Co-Exposure to Physical Sunscreen Ingredients Zinc Oxide and Titanium Dioxide Nanoparticles"

_nanomaterials, 2022, doi:10.3390/nano12162769_

Round 1
Reviewer 1 Report
The presented manuscript includes the study on skin toxicity of co-exposure to physical sunscreen ingredients TiO2 NPs and ZnO NPs and the inhibitory effect of TiO2 NPs on the cellular uptake of aggregated ZnO NPs and non-aggregated ZnO NPs. The paper is well written and well structured and the topic presents an actual interest in condition of using sun screen product on large scale. The results of the work are presented on a good level and as result recommend publication with a minor observation, namely Zinc oxide and Titanium dioxide must be written as zinc oxide (ZnO) and titanium dioxide inside the sentences.
Reviewer 2 Report
It is very interesting and well described study. Congratulations!
Reviewer 3 Report
In this paper the authors studied the antagonistic joint skin toxicity of coexposure to physical sunscreen ingredients, in particular to TiO2 NPs and ZnO NPs and analysed the inhibition effect of TiO2 NPs on the cellular uptake of aggregated ZnO NPs and non-aggregated ZnO NPs.
The results presented proposed the competitive inhibition effect of these two NPs on the caveolae-mediated endocytosis. Furthermore, they verified these antagonistic joint skin effects on the in vitro RHE models EpiSkinTM and further demonstrated that the ZnO NPs transdermal restriction induced by TiO2 NPs played an essential role in decreasing the dermal toxicity of the co-exposure system.
The paper is very interesting, well written and new.
I don't see particular criticisms.
My only comment regards the quality of the figures, several plots are hard to be read, larger text size could be useful
Reviewer 4 Report
In the present paper, it is studied the skin toxicity effects of co-exposure to physical sunscreens, using mixtures of zinc oxide and titanium dioxide nanoparticles. Both components are often used together in different sunscreen products with different proportions, and the authors revealed the antagonistic joint skin toxicity of their co-exposure and analyzed the inhibition effect of TiO2 NPs on the cellular uptake of aggregated ZnO NPs and non-aggregated ZnO NPs. This was verified in vitro with Human-derived keratinocytes (HaCaT) and using the reconstructed human epidermis (RHE) model EpiSkin TM. This manuscript provides data for explaining combined dermal toxicity and contributes for formulations on the application of ZnO NPs and TiO2 NPs in sunscreens.
General comments:
The manuscript is scientifically sound, and the experimental design is appropriate and adequate to the technical standards. It should be acknowledged the authors' priority for the use of in vitro assays, instead of in vivo treatments, for this cosmetics research. The article is written in an appropriate way (English level is appropriate and clear), being data robust and analyzes presented appropriately. Data seems to be interpreted appropriately and consistently throughout the manuscript. Conclusions are consistent with the evidence and arguments presented, being the review clear, comprehensive and of relevance to the field. The cited references are mostly recent and relevant publications (within the last 5 years)
Specific Comments:
Regulation on sunscreen components, which are nonprescription drugs, help to make sure that consumers have access to safe and effective sun protection products. Recent International regulation about UV filters concentrations, should be presented and mentioned in the reference section (few publications were presented).
Caution should be taken when new sunscreens or formulations are developed and research that includes sunscreen NPs stabilization, effects of emulsifiers, chronic exposures, etc, is essential. Could you please add some discussion about these issues?
In section 3.8, the first paragraph could be merged in section 2.11.
